# Free and Open-Source Posologyr Software for Bayesian Dose Individualization: An Extensive Validation on Simulated Data

**DOI:** 10.3390/pharmaceutics14020442

**Published:** 2022-02-18

**Authors:** Cyril Leven, Anne Coste, Camille Mané

**Affiliations:** 1Department of Biochemistry and Pharmaco-Toxicology, Brest University Hospital, 29200 Brest, France; camille.mane@chu-brest.fr; 2Univ Brest, EA 3878, GETBO, 29200 Brest, France; 3Infectious Diseases Department, Brest University Hospital, 29200 Brest, France; anne.coste@chu-brest.fr

**Keywords:** clinical pharmacokinetics, dosage individualization, Bayesian dosing, therapeutic drug monitoring, Maximum A Posteriori

## Abstract

Model-informed precision dosing is being increasingly used to improve therapeutic drug monitoring. To meet this need, several tools have been developed, but open-source software remains uncommon. Posologyr is a free and open-source R package developed to enable Bayesian individual parameter estimation and dose individualization. Before using it for clinical practice, performance validation is mandatory. The estimation functions implemented in posologyr were benchmarked against reference software products on a wide variety of models and pharmacokinetic profiles: 35 population pharmacokinetic models, with 4.000 simulated subjects by model. Maximum A Posteriori (MAP) estimates were compared to NONMEM post hoc estimates, and full posterior distributions were compared to Monolix conditional distribution estimates. The performance of MAP estimation was excellent in 98.7% of the cases. Considering the full posterior distributions of individual parameters, the bias on dosage adjustment proposals was acceptable in 97% of cases with a median bias of 0.65%. These results confirmed the ability of posologyr to serve as a basis for the development of future Bayesian dose individualization tools.

## 1. Introduction

Model-informed precision dosing (MIPD) is an emerging dosing paradigm in which mathematical models are used for dose optimization, using individual information such as a patient’s age, organ function, and the results of therapeutic drug monitoring. Driven by the increasing importance of information technology in health care [1], MIPD is being promoted as a strategy to improve the efficiency of therapeutic drug monitoring and improve standards of care [2].

Because of the complexity of generic modeling software such as NONMEM, MIPD generally requires custom-made software tools tailored to the needs of clinical providers [1]. User-friendly software such as MWPharm [3], TDMx [4], and many others [5] have been developed to meet various needs. Some of this software is freely available, some allows specialist users to integrate their own models, and some has been validated for their predictive performance, but few tools combine all of these features. Software that utilizes publicly documented, accessible and reusable algorithms is even rarer.

Moreover, most of these software products rely only on Maximum A posteriori Bayesian Estimation (MAP-BE), assuming that the mode of the posteriori distribution leads to the prediction with the highest probability. However, Bayesian inference goes beyond point estimates: full posterior (conditional) distributions retain information about the uncertainty associated with individual parameter estimates. Taking this uncertainty into account helps to improve the quality of the predictions, and full Bayesian approaches have been shown superior to MAP-based approaches in dosing individualization [6].

The posologyr software has been developed to address these issues. Featuring Bayesian inference of individual parameters based on parametric nonlinear mixed-effects modeling, posologyr is a free R package [7], licensed under AGPLv3 [8] to guarantee external auditability and accessibility to the widest possible audience of R users.

The aim is for posologyr to be a reliable and open foundation for the development of future MIPD tools. To this end, the extensive validation of the predictive performance of the various implemented functions was an essential requirement.

## 2. Materials and Methods

### 2.1. Nonlinear Mixed Effects Models

Nonlinear mixed-effects models are frequently used for the analysis of pharmacokinetic data. They take into account different levels of variability, including inter-individual variability, by incorporating random effects [9]. The observations can be described using the following general nonlinear mixed-effects model implemented in the NONMEM software:(1)$$y_{ij}=f\left(x_{ij},\psi_{i}\right)+g\left(x_{ij},\psi_{i},\varepsilon_{ij}\right)$$
for *i* from 1 to *N*, and for *j* from 1 to *n*_i_. Where,

$y_{ij}$ is the *j*^th^ observation of subject *i*;N is the number of subjects;$n_{i}$ is the number of observations of subject *i*;*f* is the function defining the structural model;*g* is the function defining the residual error model;$x_{ij}$ is the vector of regression variables;for subject *i*, the vector $\psi_{i}$ is a vector of individual parameters:
(2)$$\psi_{i}=H\left(\theta ,c_{i},\eta_{i}\;\right)$$
where,
○θ is a vector of fixed effects;○$c_{i}$ is a vector of covariates;○$\eta_{i}$ is a vector of normally distributed random effects, of length *k*, with variance-covariance matrix Ω:



(3)
$$\eta_{i}\;\sim \;N\left(0,\Omega \right)$$



The residual errors $\varepsilon_{ij}$ are normally distributed random variables centered on 0, with variance Σ:



(4)
$$\varepsilon_{ij}\;\sim \;N\left(0,\Sigma \right)$$



In the Monolix software, the implementation of the residual error model differs, and the nonlinear mixed-effects model becomes:(5)$$y_{ij}=f\left(x_{ij},\psi_{i}\right)+g\left(x_{ij},\psi_{i},\;\xi \right)\varepsilon_{ij}$$
where the residual error model is defined by the function *g* and some parameters ξ. The residual errors are then random variables with mean zero and variance one.

### 2.2. Estimation of Individual Parameters

#### 2.2.1. General Strategy

In the context of MIPD, the aim is to determine the individual parameters $\psi_{i}$ of a subject *i*, for which dosage personalization is needed. The estimation of individual parameters requires (i) a model for (y,ψ) with the prior estimates of θ, $x_{ij}$, the Ω  variance covariance matrix, and, depending on the definition of the residual error model, either the Σ matrix, or the vector of parameters ξ; (ii) the input of the observed concentrations y, the measurement times t, the individual covariates, and the administration regimen; and (iii) algorithms capable of estimating and maximizing $\;p(\psi_{i}|y_{i})$. The fixed effects of the population pharmacokinetic model are used as prior information, and the values of the individual random effects $\eta_{i}$ are estimated a posteriori, taking advantage of the observed data.

#### 2.2.2. Maximum A Posteriori

MAP estimation determines the vector of individual parameters with the highest probability, i.e., the mode of the posterior distribution. In NONMEM, it is performed during the POSTHOC step by minimizing the following objective function value (OFV), based on the log likelihood (LL) [10]:(6)$${\mathrm{OFV}}_{i}=-2\mathrm{LL}\left(\eta_{i}\right)=\sum_{j}\left[\log\sigma^{2}{}_{ij}+\frac{{\left(y_{ij}-f_{ij}\right)}^{2}}{\sigma^{2}{}_{ij}}\right]+\eta_{i}{}^{T}\Omega^{-1}\eta_{i}$$
where $\sigma^{2}{}_{ij}$ is the variance of the residual error for individual *i* at time *j*.

While most Bayesian forecasting tools rely on the MAP estimation, this approach does not necessarily predict the most probable outcome [6]. To provide a comprehensive uncertainty quantification, it may be desirable to compute the full posterior distributions, and not just point estimates such as the MAP.

#### 2.2.3. Markov Chain Monte Carlo

While the probabilities of $p\left(\psi_{i}\right)$ or $\;p\left(\eta_{i}\right)$ cannot be directly calculated, Markov chain Monte Carlo (MCMC) methods allow this distribution to be sampled. The typical procedure is as follows: at each iteration of the algorithm, a new vector of individual parameters $\psi_{i}$ is drawn from a proposal distribution. The new value is accepted with a probability that depends on $p\left(\psi_{i}\right)$ and on $\;p(y_{i}|\psi_{i})$. After a transition period, the algorithm reaches a stationary state where the accepted samples come from the posterior probability distribution $\;p(\psi_{i}|y_{i})$.

#### 2.2.4. Sequential Importance Resampling

Particle filter algorithms, including Sequential Importance Resampling (SIR), are another class of algorithms that asymptotically draw samples from the posterior probability distribution, allowing for estimation of $\;p(\psi_{i}|y_{i})$. The SIR algorithm, consists of 3 steps [11]:Step 1 (sampling): a defined number *M* of parameters are sampled from a multivariate parametric proposal distribution;Step 2 (importance weighting): weights are computed for each of the sampled vectors, using the likelihood of the data given the parameter vector, weighted by the likelihood of the parameter vector in the proposal distribution;Step 3 (resampling): *m* parameter vectors are resampled from the *M* simulated vectors (*M* > *m*), with probabilities proportional to their weighting.

Each step of the algorithm can be parallelized: for the same number of samples, the time needed for estimation can be significantly reduced when compared to MCMC.

### 2.3. Implementation in Posologyr

The posologyr package (available at https://github.com/levenc/posologyr/, accessed on 15 January 2022) builds on the RxODE [12] simulation framework, using the Fortran package LSODA (Livermore Solver for Ordinary Differential Equations) for solving systems of differential equations. It does not depend on any non-free licensed software, such as NONMEM or Monolix, to run. Random effects of individual models (inter-individual, inter-occasion), and residual error models are implemented within posologyr; RxODE is employed as a deterministic simulation engine. For the MAP estimation, the OFV minimization is performed with the L-BFGS-B algorithm [13] included in the optim package, a limited-memory approximation of the Broyden–Fletcher–Goldfarb–Shanno algorithm (BFGS) allowing for box constraints. To avoid converging to a suboptimal local minimum of the OFV, initial values of the parameters are drawn from the multinormal distribution $\;\eta_{i}\;\sim \;N\left(0,\Omega \right)$. The MCMC algorithm of posologyr is an adaptation of the Metropolis–Hastings algorithm from the R package saemix [14], slightly modified to estimate the posterior distribution $\;p\left(\eta_{i}|y_{i}\right)$ and to use RxODE as a simulation engine. As in the Metropolis–Hastings algorithm featured in Monolix, three different distributions are used in turn with a (2,2,2) pattern for the proposal distribution: the population distribution, a unidimensional Gaussian random walk, or a multidimensional Gaussian random walk [15]. For the random walks, the variance of the Gaussian is automatically adapted to reach an optimal acceptance ratio. The number of iterations to be discarded following the burn-in period, the number of Markov chains, and the number of iterations per chain are user-defined. The SIR algorithm implemented in posologyr works by sampling from the multinormal distribution  N(0,Ω) using the rmvnorm function of the mvtnorm package. The weightings are computed from the LL, i.e.,−0,5×OFV, following the simultaneous simulation of the observations generated by the *M* individual parameter vectors from the sampling step. The initial number of samples *M*, and the number of draws *m* in the resampling step can be defined by the user.

### 2.4. Dosing Adjustment

In order to individualize treatments, posologyr features several dosage optimization functions based on individual parameter estimates. The posologyr::poso_dose_conc and posologyr::poso_dose_auc functions determine the optimal dose to reach, respectively, a target concentration at a given time, or a target area under the time-concentration curve (AUC) over a given duration. The optimization is based on the minimization of the square of the difference between the desired target (target concentration, target AUC) and the result of the successive simulations. The posologyr::poso_time_cmin function determines by simulation the time needed to reach a target concentration (typical application: the therapeutic drug monitoring of aminoglycoside treatment). Finally, the posology::poso_inter_cmin function estimates the optimal inter-dose interval to reliably achieve a target trough concentration between each administration. All these functions allow optimization from MAP estimates of individual parameters, but also from posterior distributions. In the latter case, the set of probable profiles is simulated using RxODE, and the proportion *p* of the distribution to be considered for optimization is set by the user. For example, for posologyr::poso_dose_auc with *p* = 0.5, posologyr will determine the optimal dose so that 50% of the probable profiles reach or exceed the target AUC.

### 2.5. Validation

#### 2.5.1. Point Estimate: MAP

To allow for an extensive comparison of the performance of posologyr estimates and NONMEM POSTHOC estimates [16] (NONMEM version 7.4.4, Dublin, Ireland: ICON), the validation of the posologyr MAP algorithm was performed following the methodology proposed by Le Louedec et al. [17]. The 35 population pharmacokinetic models (Table 1) were transcribed for posologyr (an example is given in Appendix A): default monocompartmental with linear elimination, bicompartmental, with various absorption models (lag-time, zero order, first order, combination of zero and first order kinetics, bioavailability), with nonlinear Michaelis–Menten elimination associated or not with linear elimination, with time-varying covariates, different residual error models (additive, proportional, mixed, log-additive), with two types of observations (parent–metabolite model), and finally with increasing levels of inter-individual variability (with variances ranging from 0.2 to 2).

For models with time-varying covariates, the interpolation of variables between observations was performed using the “next observation carried backward” (nocb) approach, similar to NONMEM. The data set simulated by mrgsolve [18] included 4000 subjects per model, or 140,000 subjects in total. They were equally divided into 4 administration and sampling schemes: single administration and rich sampling, single administration and sparse sampling, multiple administration and rich sampling, and multiple administration and sparse sampling. Doses of 10, 30, 60, 80, or 120 mg were administered to each group. To address the nondeterministic nature of the selection of initial $\eta_{i}$ for OFV minimization, the pseudorandom generator seed was set to an identical value for all estimates using the set.seed function of R.

#### 2.5.2. Conditional Distributions

The estimates of the posologyr MCMC and SIR algorithms were compared to the estimates of the MCMC algorithm from the conditional distributions task of Monolix (version 2021R1. Antony, France: Lixoft SAS, 2020). As the computational time per individual was significantly higher than for the validation of the MAP algorithm, the validation was performed on a subset of the test models and the simulated subjects. The conditional distributions were estimated based on 10,000 samples of each algorithm, for each parameter. The SIR algorithm was run with settings *M =* 10^6^ samples, and with *m =* 10^4^ resampling runs. The MCMC algorithm of posologyr was run with 2 Markov chains, each starting with 200 burn-in iterations, discarded before drawing 5000 samples from each chain. The models tested were as follows: the default single-compartment model with oral administration (model #1); the model with lag-time (#3); the model with dual zero and first order absorption (#6); the bicompartmental model with oral administration (#101); the model with nonlinear Michaelis–Menten elimination and oral administration (#201); the model with both linear and nonlinear Michaelis–Menten elimination with oral administration (#207); the model with time-dependent clearance and oral administration (#301); the model with mixed residual error (#405); the model with variance of the random effects set to 1 for all parameters (#504); and the model with variance of random effects equal to 2 (#513). For the model with time-varying covariates, the interpolation of variables between observations was performed using the “last observation carried forward” (locf) approach, similar to Monolix [19]. For model #405, the mixed residual error model was set to combined1 in Monolix. To test the different administration and sampling schemes, 20 subjects were estimated for each model (to test each administration/sampling cohort, and every dose level), i.e., 200,000 parameter samples estimated per algorithm. To ensure reproducibility of the experiments, the pseudorandom generator seed was set to an identical value for all estimates using the set.seed function of R for posologyr, and in the project settings for Monolix.

### 2.6. Performance Analysis

#### 2.6.1. Point Estimate: MAP

To allow comparability of the results of the performance evaluation of the MAP algorithm of posologyr, the primary endpoints were identical to those proposed by Le Louedec et al. [17]. The maximum absolute difference was obtained between posologyr ($\eta_{ik,\;PGYR}$) and NONMEM ($\eta_{ik,NM}$) for each individual according to the following expressions:(7)$$\Delta {\hat{\eta }}_{ik}=\left|{\hat{\eta }}_{ik,PGYR}-{\hat{\eta }}_{ik,NM}\right|$$
(8)$$\Delta {\hat{\eta }}_{i}=\max\left(\Delta {\hat{\eta }}_{ik}\right)$$
where *k* is the length of the vector ${\hat{\eta }}_{i}$. Based on this definition, several performance thresholds were defined. For $\Delta {\hat{\eta }}_{i}$ < 0.001, the estimate was considered excellent because, assuming an individual exponential model, with a lognormal distribution of individual parameters around a population median value, the impact on the parameter estimate would be negligible: $\Delta \psi_{i}$ < 0.1%. For $\Delta {\hat{\eta }}_{i}$ > 0.095, the estimate was considered discordant ($\Delta \psi_{i}$ > 10%). The other estimates were considered acceptable. Finally, for each estimate, the final result of the calculation of the OFV from the MAP estimates of posologyr was compared to the OFV computed from the NONMEM POSTHOC estimates.

#### 2.6.2. Conditional Distributions

Because of the stochastic nature of the estimation algorithms, and because of the multivariate nature of the individual parameter vectors, the isolated posterior distributions were not directly compared using numerical criteria. The validation endpoints were the bias between the results of three different dosage adjustment functions built into posologyr, taking the optimizations based on the posterior distributions produced using Monolix as the reference. For each subject, and each function, the bias was calculated as follows:(9)$$bias=\left|\frac{Output_{algorithm}-Output_{Monolix}}{Output_{Monolix}}\right|$$
where $Output_{Monolix}$ is the result of the dose adjustment function based on the distribution of individual parameters estimated using Monolix, and $Output_{algorithm}$ is the result of the same dose adjustment function based on the distribution of individual parameters estimated using either MCMC or SIR. One scenario per function was tested:

Determination of the optimal dose to reach a concentration of 30 mg/L, 3 h after administration, with posologyr::poso_dose_conc;Determination of the optimal dose to achieve an AUC0-12h of 500 mg·h/L, with posologyr::poso_dose_auc;Determination of the time to reach a trough concentration below 0.5 mg/L after a 100 mg dose, with posologyr::poso_time_cmin.

The *p* proportion was set at 0.89 in all cases. For all outcomes, the bias was considered acceptable when ≤10%.

## 3. Results

### 3.1. Point Estimate: MAP

The performance of the posologyr MAP algorithm was satisfactory. The estimates were excellent in 98.7% of the cases. The median $\Delta {\hat{\eta }}_{i}$ was 7.70 × 10^−6^, and only 0.58% of all estimates were discordant (Figure 1). Representative pharmacokinetic profiles of excellent and discordant estimates, compared with NONMEM profiles, are given in Appendix B (Figure A1).

The greatest number of discordant estimates was observed for the models with lag-time (model #3), and those with zero order absorption in the central compartment (models #4 and #6). For these models, no discrepancies were found in the case of sparse sampling. In contrast, for the 6000 patients in models 3, 4, and 6 with rich sampling, 9.5% of the estimates were discordant. In the majority of these cases (402 of 570 subjects, or 70% of the estimates), the OFV associated with the posologyr estimate was lower than that calculated based on the NONMEM POSTHOC estimate, in support of a better performance of minimization when using posologyr, leading to the most likely value of ${\hat{\eta }}_{i}$. Discordant MAP estimates were also more frequently observed in models with increasing inter-individual variability (models #504, #511, #512 and #513), up to a maximum of 3.3% for model 513, irrespective of the administration or observation scheme. Again, for model #513, the posologyr OFVs were lower than the NONMEM POSTHOC estimate in most cases (24 subjects out of 4000, or 99.4%). Extending the above observation regarding OFV to all 140,000 simulated subjects, the estimates produced using posologyr were outperformed by the NONMEM POSTHOC estimate in only 0.14% of cases.

### 3.2. Posterior Distribution

The performance of the SIR algorithm was satisfactory. Using the posologyr::poso_dose_conc and posologyr::poso_dose_auc functions, the bias on dosage adjustment proposals based on SIR estimates was acceptable in 97% of cases (Figure 2 and Figure 3). The median bias was 0.65% (minimum: 0.00%, maximum: 31.41%). Elimination time predictions using posologyr::poso_time_cmin were acceptable in 99.5% of cases (all but one subject) and the median bias was 0.14% (minimum: 0.00%, maximum: 23.16%).

All outcomes considered, the greatest number of discordances between SIR and Monolix was observed for models with significant inter-individual variability (#504 and #513): 8.3%. For these subjects, proposals based on estimates from the MCMC algorithm implemented in posologyr were discordant with propositions based on Monolix estimates in 10.8% of cases. For the entire dataset, the estimates produced using MCMC exhibited 4.17% discrepancies with Monolix, with a median bias of 0.77%.

For illustration purposes, the density curves of the posterior distributions computed using the different algorithms for subject #1, model #003 are presented in Figure 4. 

## 4. Discussion

A free and open-source software has been developed to allow for the Bayesian individualization of treatments. The performance of posologyr for individual parameter estimation was benchmarked against NONMEM for MAP point estimates and against Monolix for the estimation of full posterior distributions of individual parameters.

The MAP optimization function of posologyr showed excellent performance compared to NONMEM for a wide variety of models and pharmacokinetic profiles. In particular, agreement with NONMEM’s POSTHOC estimates was excellent for the single-compartment and two-compartment models, the models with nonlinear Michaelis–Menten elimination, the model with time-dependent clearance, and the various residual error models, for values of inter-individual variability lower than 1.

A greater number of discrepancies was observed for the different absorption models, especially following administration with lag-time. This type of model produces discontinuous time-concentration profiles, which to some degree may be responsible for difficulties in determining the optimal values of individual parameters [20]. An increasing, but small, number of discrepancies was also observed for increasing values of inter-individual variability. In these different cases, under the assumption that the OFV calculated from the estimated individual parameters [10] is a satisfactory reflection of the minimization performance, posologyr outperformed NONMEM’s POSTHOC estimates for the majority of discordances.

The SIR algorithm implemented in posologyr demonstrated satisfactory performance on the evaluated scenarios. The estimates of the posterior distributions of individual parameters led to dose adjustment proposals comparable to those based on the Monolix estimates. The execution time of the different algorithms was not formally evaluated; however, the observations made during this study showed that the time required to obtain 10,000 samples was 15 times shorter with the SIR algorithm of posologyr than with the MCMC algorithm of posologyr. These results are reassuring with regard to the feasibility of developing a probabilistic MIPD tool in R, opening the door to personalized treatment options informed by knowledge of the uncertainty associated with individual parameter estimation [6]. Still, in this preliminary study, the simulated population tested was limited to 200 patients, with one scenario per function, and a single probability threshold; these exploratory results need to be confirmed before considering the application of this methodology to therapeutic drug monitoring. A thorough evaluation of the performance of SIR will be necessary in order to define the precise scope of its application according to model types, sampling schemes, and patient profiles. The evaluation of different sample sizes, and resampling numbers, would also allow for the optimization of runtime without foregoing estimation quality.

The posologyr software is, and will remain, free and open source under the AGPLv3 license [8]. Now available at https://github.com/levenc/posologyr (accessed on 15 January 2022), it is scheduled for release via the comprehensive R archive network (CRAN) soon. These decisions to make the software easy to distribute and to guarantee its external auditability serve the purpose for which posologyr was developed: to provide a foundation for future Bayesian dosage individualization tools.

## Figures and Tables

**Figure 1 pharmaceutics-14-00442-f001:**
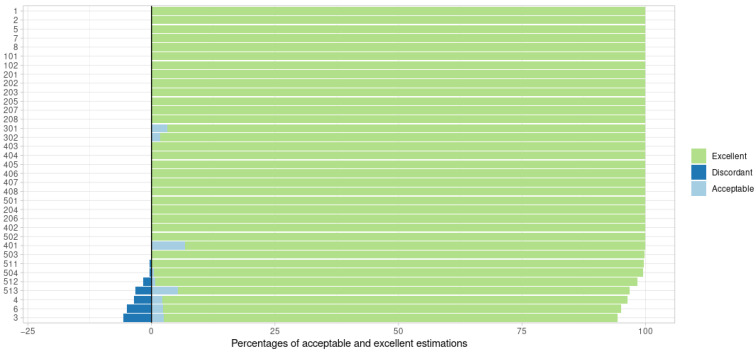
Performance of posologyr MAP estimation by test model. The negative percentages are the proportions of discordant estimates.

**Figure 2 pharmaceutics-14-00442-f002:**
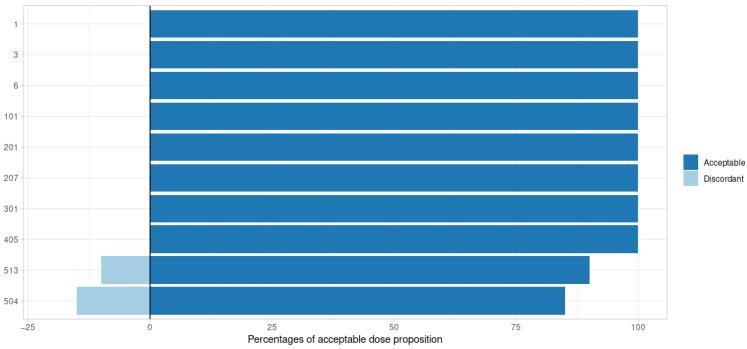
Adequacy of dose adaptation by tested model for a target AUC based on SIR posterior estimates. The negative percentages are the proportions of discordant propositions.

**Figure 3 pharmaceutics-14-00442-f003:**
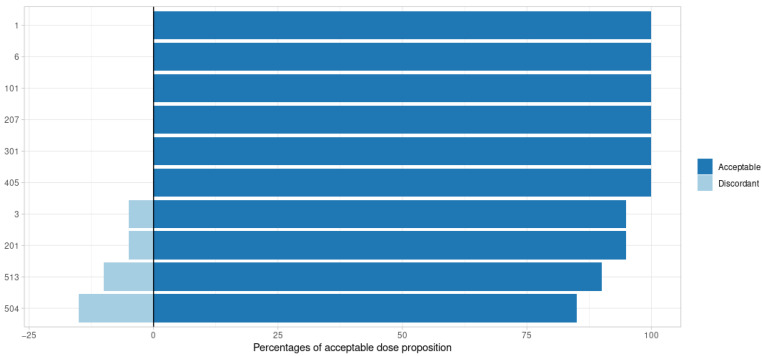
Adequacy of dose adaptation by tested model for a target concentration based on SIR posterior estimates. The negative percentages are the proportions of discordant propositions.

**Figure 4 pharmaceutics-14-00442-f004:**
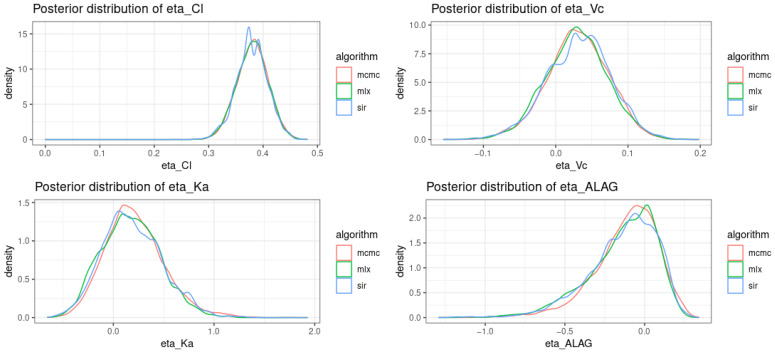
Density plots of the posterior distributions of $\eta_{i}$ for subject #1 with model #003.mcmc, the MCMC algorithm implemented in posologyr; mlx, the MCMC algorithm of Monolix; sir, the SIR algorithm of posologyr; Cl, clearance; Ka, absorption rate; Vc, central volume of distribution; ALAG, lag-time.

**Table 1 pharmaceutics-14-00442-t001:** Test models.

**Model Characteristics**		Model Number (Number of Estimated Parameters)	
		Oral Administration	IV Administration
Monocompartmental (default)		1 (3)	2 (2)
Absorption	Lag time	3 (4)	/
	Zero-order in Central compartment	4 (3)	/
	Zero-order in Depot compartment	5 (4)	/
	Dual 0- and 1st orders	6 (4)	/
	Dual 1st orders	7 (4)	/
	Bioavailability	8 (4)	/
Distribution	Bicompartmental	101 (4)	102 (3)
Elimination	Michaelis–Menten (K_M_, V_MAX_)	201 (4)	202 (3)
	Cl + Michaelis–Menten (K_M_)	203 (4)	204 (3)
	Cl + Michaelis–Menten (V_MAX_)	205 (4)	206 (3)
	Cl + Michaelis–Menten (K_M_, V_MAX_)	207 (5)	208 (4)
Time-Varying Covariates	Time-varying Cl	301 (3)	302 (2)
Residual Error Model	Metabolite	401 (5)	402 (4)
	Additive	403 (3)	404 (2)
	Mixed	405 (3)	406 (2)
	Log-additive	407 (3)	408 (2)
Inter-individual Variability (variance)	0.4 on all parameters	501 (3)	/
	0.6 on all parameters	502 (3)	/
	0.8 on all parameters	503 (3)	/
	1 on all parameters	504 (3)	/
	2 on Cl, 0.2 on Ka, Vc	511 (3)	/
	2 on Cl, Ka, 0.2 on Vc	512 (3)	/
	2 on all parameters	513 (3)	/

Abbreviations: Cl, clearance; IV, intravenous 1 h infusion; Ka, absorption rate; KM, Michaelis–Menten constant; Vc, central volume of distribution; VMAX, maximum rate; /, not applicable.

## Data Availability

Data supporting the reported results can be found at https://github.com/levenc/posologyr-pharmaceutics (accessed on 15 January 2022).

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
