# Peer review of "Free and Open-Source Posologyr Software for Bayesian Dose Individualization: An Extensive Validation on Simulated Data"

_pharmaceutics, 2022, doi:10.3390/pharmaceutics14020442_

Round 1
Reviewer 1 Report
pharmaceutics-1578516
Free and open source posologyr software for Bayesian dose in3 dividualization: an extensive validation on simulated data
The features of a software posologyr are reviewed and its advantages are shown in calculation of a model-informed precision dosing in order to improve therapeutic drug monitoring. It has been concluded that posologyr can be used as a basis for development of future Bayesian dose individualization tools. Very big advantage of the article is clear stile of the presentation of the subject.
Abstract is enough informative as well. The key words are suitable.
Materials and Methods
This section explains the general principles of nonlinear mixed effects. The description is very clear and easy for understanding even from the beginners. This is very important because one of the aim of the manuscript is to show the ability of wide usage of software posologyr.
Results
All the calculations that were performed with different models, algorithms and software are well presented. The figures help to understanding the observed similarities/discrepancies.
Discussion
Discussion properly reflect the obtained results. Further requirements for the validation of the used algorithm in dose optimization are outlined.
In conclusion: The manuscript is well written and clear. It presents for the first time a possibility to apply SIR algorithm in optimization of the dose by usage of software posologyr. It deserves to be published. The only remark is related to a possibility to show in a table, as a supplementary material, the obtained results from the calculated doses for the models that gave similar results and for the models that gave quite different (non-acceptable) results when posologyr software was compared to Monolix.
Author Response
Since the models are fictitious, and the drugs are fictitious, the proposed dose crude values may not be informative. However, we feel that the point raised by reviewer #1 is important.
To provide more information on the discrepancies between the dose proposals based on posologyr estimates, and those of Monolix, the minimum and maximum biases have been added to the manuscript. In section 3.2., the following sentences:
The median bias was 0.65%. Elimination time predictions using posologyr::poso_time_cmin were acceptable in 99.5% of cases (all but one subject), the median bias was 0.14%.
have been replaced by the following:
The median bias was 0.65% (minimum: 0.00%, maximum: 31.41%). Elimination time predictions using posologyr::poso_time_cmin were acceptable in 99.5% of cases (all but one subject), the median bias was 0.14% (minimum: 0.00%, maximum: 23.16%).
Reviewer 2 Report
The aim of optimal pharmacotherapy is to tailor the medical treatment to the individual characteristics of each patient. In this regard, the results presented by the authors are relevant and can be used as recommendations for clinical use.
English language is not quite at the publication level. Minor errors are related to the articles and to the verb "to be".
Author Response
Following the recommendation of reviewer #2, the manuscript was proofread to correct English language errors.
Reviewer 3 Report
In this manuscript, the authors present open-source software created in the R language that can be used for dose individualization.
The authors evaluate the performance of the new software and, in this context, present the validation results against NONMEM. The latter show good performance.
It is an interesting article that responds to the need for open-source software, especially in the hospital sector.
Comment:
For completeness, the authors may add in the Appendix some snapshots of the diagrams produced in the simulations of the dosing regimens.
Author Response
As suggested, illustrations of the simulated pharmacokinetic profiles from the MAP estimates of posologyr and NONMEM, and for different dose profiles, have been added in Appendix.